# The Effect of Nutritional Intervention with Lactoferrin, Galactooligosacharides and Vitamin D on the Gut Microbiota Composition of Healthy Elderly Women

**DOI:** 10.3390/nu14122468

**Published:** 2022-06-14

**Authors:** Prokopis Konstanti, Marloes van Splunter, Erik van den Brink, Clara Belzer, Arjen Nauta, R. J. Joost van Neerven, Hauke Smidt

**Affiliations:** 1Laboratory of Microbiology, Wageningen University & Research, 6700 EH Wageningen, The Netherlands; clara.belzer@wur.nl (C.B.); hauke.smidt@wur.nl (H.S.); 2Cell Biology & Immunology, Wageningen University & Research, 6700 AH Wageningen, The Netherlands; marloes.vansplunter-berg@wur.nl (M.v.S.); erik.vandenbrink@wur.nl (E.v.d.B.); joost.vanneerven@wur.nl (R.J.J.v.N.); 3Aquaculture and Fisheries, Wageningen University & Research, 6700 AH Wageningen, The Netherlands; arjen.nauta@frieslandcampina.com; 4FrieslandCampina, 3818 LA Amersfoort, The Netherlands

**Keywords:** bovine lactoferrin, galactooligosaccharides, gut microbiota, elderly

## Abstract

Background: Nutritional supplements, such as bovine lactoferrin (bLF), have been studied for their immunomodulatory properties, but little is known of their effect on the gut microbiota composition of the elderly when supplemented alone or combined with other nutritional supplements such as prebiotics and micronutrients. In the present study, fecal samples from a double-blind, placebo-controlled nutritional intervention study were analysed. At baseline (T1), 25 elderly women were distributed into two groups receiving dietary intervention (*n* = 12) or placebo treatment (*n* = 13) for 9 weeks. During the first 3 weeks of the study (T2), the intervention group consumed 1 g/day bLF, followed by 3 weeks (T3) of 1 g/day bLF and 2.64 g/day active galactooligosaccharides (GOS), and 3 weeks (T4) of 1 g/day bLF, 2.64 g/day GOS and 20 μg/day of vitamin D. The placebo group received maltodextrin, in dosages matching those of the intervention group. Fecal bacterial composition was profiled using partial 16S rRNA gene amplicon sequencing. Short-chain fatty acids (SCFA) were determined in fecal water as were levels of calprotectin, zonulin, and alpha-1-antitrypsin, as markers of gastrointestinal barrier and inflammation. Results: A significant increase was observed in the relative abundance of the genus *Holdemanella* (*p* < 0.01) in the intervention group compared to the placebo at T1. During T2, *Bifidobacterium* relative abundance increased significantly (*p* < 0.01) in the intervention group compared to the placebo, and remained significantly higher until the end of the study. No other effect was reported during T3. Furthermore, concentrations of SCFAs and calprotectin, zonulin and alpha-1-antitrypsin did not change during the intervention, although zonulin levels increased significantly within the placebo group by the end of the intervention. Conclusions: We conclude that supplementation of bLF enhanced the relative abundance of *Holdemanella* in the fecal microbiota of healthy elderly women, and further addition of GOS enhanced the relative abundance of *Bifidobacterium*.

## 1. Introduction

Aging is associated with a decline in immune system functioning known as immuno-senescence, characterized by poor pathogen clearance, and low-grade chronic inflammation [1], making the elderly population more susceptible to infections and to age-related comorbidities [2,3]. With the worldwide elderly population increasing, especially in most Western countries, managing the healthcare system is a major challenge for future generations [4]. Hence solutions are necessary to support immune function and promote healthy aging in our societies.

In recent years, dietary interventions with immunomodulatory supplements were explored as an alternative to pharmacological interventions [5]. Among others, bovine Lactoferrin (bLF) has attracted attention, due to its immuno-modulatory [6,7], antimicrobial [8], antiviral [9], and antioxidative abilities [10,11]. In vitro studies showed that bLF has the potential to suppress the growth of enteric pathogenic bacteria [12], through iron scavenging [13] or through antimicrobial peptides that derive from proteolytic degradation of bLF [14]. Furthermore, bLF inhibits viral attachment and replication through the modulation of the immune system or through the binding with heparan sulfate proteoglycans, which blocks the entry of virus to the host cell and prevents infections [15]. These characteristics make bLF a relevant candidate as a nutritional supplement for the elderly, but in vivo studies, in this age group are limited [16].

Despite the evidence that bLF can modulate the immune system and affect pathogen survival, little is known of its effect on the gut microbiota composition when administrated to the elderly. Gut microbiota composition and metabolic activity is increasingly acknowledged as an important component of gastrointestinal homeostasis [17] and microbial disbalances have been associated with the progression of disease during aging [18]. Several studies identified differences in the gut microbiota composition of elderly and young adults, [19] the metagenomic content [20], carbohydrate degradation and metabolite production capacity of the elderly in in vitro experiments [21]. Overall, compared to younger adults, the gut microbiota in the elderly has frequently been reported to contain lower levels of bifidobacteria and higher levels of potential pathogens, such as members of the Enterobacteriaceae family [18]. Bifidobacteria play an important role in gut health throughout life [22], and nutritional interventions with prebiotics such as galactooligosaccharides (GOS) have been shown to promote bifidobacteria in the elderly and modulate the immune system through the reduction of the levels of circulating pro-inflammatory cytokines [23].

Previously, we have shown that bLF can enhance TLR-7 responses in plasmacytoid dendritic cells, suggesting that bLF may support antiviral immunity to single-stranded RNA viruses [7]. Currently, in the literature only one small pilot study has examined the effect of bLF on the gut microbiota composition of participants, with minor effects recorded [24]. In our previous study, we focused on the effect of staged supplementation of bLF, followed by addition of GOS and vitamin D, and determined immune parameters in elderly women [7]. In the current study we aimed to characterize the effects of the intervention on gut microbiota composition and function and explored the effect on gut health based on intestinal inflammatory and barrier markers.

## 2. Materials and Methods

### 2.1. Study Design

The effect of bLF in combination with GOS and vitamin D in elderly females (65–85 years) was studied in a double-blind placebo-controlled nutritional intervention study. The protocol was approved by the Medical Ethics Committee of Wageningen University, the Netherlands (protocol no. NL57345.081.16), registered at clinicaltrials.gov (identifier NCT03026244) and was published previously [7]. Briefly, female subjects (65–85 years) were recruited with a BMI of 20–30, good mental status, non-smoking, and following regular and normal Dutch eating habits, as assessed by the NIZO lifestyle and health questionnaire which corresponds to three main meals per day. Exclusion criteria were chronic inflammatory, autoimmune, or gastrointestinal diseases or compromised immune system and medication with hormone replacement therapy, anti-inflammatory drugs (>1 × week), immunosuppressive drugs, consumption of more than three units of alcohol per day and use of prebiotics within two months prior to the start of the study.

Stratification and randomization were performed by a non-blinded person not involved in the study, and all investigators were blinded until all data were collected. Women were stratified according to age, BMI, reported arthrosis, use of vitamin D supplements preceding the study, and use of medication for blood pressure or cholesterol. Subjects were randomly assigned to treatment or placebo using a random number generator. See Table 1 for subject characteristics.

Women (*n* = 15) in the nutritional intervention group were supplemented for 3 weeks with bLF (1.026 g/day Vivinal Lactoferrin powder, containing an active dose of bLF of 1 g/day; FrieslandCampina Ingredients, Amersfoort, The Netherlands), followed by 3 weeks bLF + GOS (1 g/day bLF as above; 3.67 g/day Biotis^TM^ GOS powder, containing an active dose of GOS of 2.64 g/day; FrieslandCampina Ingredients, Amersfoort, The Netherlands), followed by 3 weeks bLF + GOS + vitamin D (Supra D Forte Supradyn, Berlin, Germany) contained 20 μg cholecalciferol (=800 IU) per capsule). In parallel, the placebo group (*n* = 15) received maltodextrin (Glucidex, IT19 premium, Roquette, Nord-Pas-de-Calais, France), as placebo for bLF and GOS, whereas maltodextrin capsules were provided as placebo for the weeks 6–9 when vitamin D was supplemented to the intervention group. Subjects were instructed to maintain their habitual diet, but to stop any vitamin D or prebiotic supplementation during the study and for 2 weeks before the beginning of the study. Fecal samples were collected by subjects at home at four time points, namely before the start of the intervention (T1), and at the end of week 3 (T2), week 6 (T3) and week 9 (T4), using a Fecotainer stool collection device (AT Medical BV, Enschede, The Netherlands) and were stored frozen at home until the end of the study, when they were transported to NIZO for storage at −20 °C. Two participants from the intervention group did not follow the collection protocol and their samples were excluded from downstream analysis. Moreover, use of antibiotics was considered as exclusion criterium, which resulted in exclusion of two participants from the placebo group and one from the intervention group.

### 2.2. Fecal Microbiota Profiling

The V5-V6 region of the 16S rRNA gene was sequenced to profile fecal microbiota composition using Illumina Hiseq2500 technology, following the same procedures as described previously [25]. Briefly, DNA was isolated from the fecal samples using repeated Beat-Beating and purified using the Maxwell^®^ 16 Instrument (Promega, Leiden, The Netherlands). The V5–V6 region (F784-R1064) of the 16S rRNA gene was amplified in duplicate PCR reactions, as described previously [25]. Subsequently, the duplicate PCR amplicons were pooled for each sample, purified with the CleanPCR kit (CleanNA, The Netherlands), and quantified using the Qubit dsDNA BR Assay kit (Invitrogen, Thermo Fisher Scientific, Eugene, OR, USA). In total, we obtained 16S rRNA gene amplicons sequences for 100 fecal samples (four samples for the 25 subjects included in the study), one PCR negative control which included water as DNA template and two artificial mock communities. Negative controls were included to identify potential contaminants in our dataset which may have been introduced during the laboratory preparation, and none of the identified taxa were present in our biological samples. The mock communities were artificially assembled communities of known 16S rRNA genes, which were included as a positive control for sequencing procedures.

### 2.3. Microbiota Data Processing and Analysis

Data filtering and taxonomy assignment were performed using the NG-Tax pipeline [26] with default parameters. The sequence read counts were normalized to microbial relative abundance and were used for α- and β-diversity analyses which were performed using the publicly available Microbiome R package [27]. ADONIS permutational multivariate analyses of variance (PERMANOVA) using the weighted UniFrac distances were calculated using the Vegan package [28] and were used to determine the amount of variation explained by the grouping variables, within each timepoint. Intervention effects between the two groups were analysed with the Mann-Whitney test comparing the changes in relative abundance (delta) between the consecutive timepoints. Paired Wilcox test was calculated with the function wilcox.exact from the exactRankTests package, and was used to compare the bacterial composition of the subjects within each group between timepoints. For the Wilcoxon analyses, 54 bacterial genera were tested after filtering using the function core from the Microbiome package [27] selecting the taxa that were present in at least 50% of the samples with relative abundances higher than 0.001%.

Wilcoxon test was applied to determine whether alpha diversity indexes (phylogenetic diversity and Shannon indexes) were significantly different between groups using a linear mixed model from the lme package [29].

### 2.4. Fecal Short Chain Fatty Acid Analysis

Fecal SCFAs were measured as described previously [30]. Standard solutions of acetic acid, lactic acid, propionic acid, butyric acid, and iso-butyric acid were used for the creation of the standard curves. Two hundred and fifty microliters of internal standard solution (0.45 mg/mL 2-ethylbutyric acid in 0.3 M HCl and 0.9 M oxalic acid) was added to 500 µL of the standard solutions and centrifuged samples. After mixing and centrifugation, 150 µL supernatant was used for analysis of SCFAs with a Thermo Scientific Spectrasystem high-performance liquid chromatography (HPLC) system equipped with a Varina Metacarb 67H 300 × 6.5 mm column kept at 45 °C and 0.005 mM sulfuric acid as eluent.

### 2.5. Measurement of Fecal Calprotectin, α1-Antitrypsin, and Zonulin

Fecal water was prepared as described above, and the proinflammatory markers calprotectin and α1-antitrypsin, as well as the intestinal barrier marker zonulin, were measured by ELISA. The ELISAs were performed as indicated by the suppliers. Human calprotectin (Human S100A8/S100A9) and human α1-antitrypsin (SERPIN A1) ELISA (DY-8226-05 and DY-1268 respectively) were purchased from R&D systems, and the human zonulin ELISA from Alere YY5600.

Fecal markers and SCFAs measured during the study were analysed using linear mixed effect models as implemented in the lme package [29], where subject identity was used as random intercept to account for the longitudinal nature of our data.

## 3. Results

In the present study, we analysed fecal samples from 25 elderly women that completed the study and did not use any antibiotics during it. Of these subjects, 12 were included in the intervention group, and 13 in the placebo group. The baseline characteristics of the two groups can be found in Table 1.

## 4. Gut Microbiota Composition between the Groups

We first compared overall microbiota composition including samples from both groups and all timepoints. Principle coordinate analysis (PCoA) based on the pairwise Bray-Curtis dissimilarity index and weighted UniFrac distance matrix did not show significant differences in the microbiota composition between the placebo and the intervention group (Appendix A) and subject identity was the variable explaining the biggest part of the variation (ADONIS; R^2^:0.654 *p* = 0.001). PERMANOVA analysis within each timepoint did not reveal any significant difference between the placebo and the intervention group (Figure 1), and neither did any of the environmental variables tested contribute significantly to explaining the observed variance (Table 2). No significant differences were detected between placebo and intervention group or within each group over time for alpha diversity, as determined by Faith’s phylogenetic diversity and Shannon index (Appendix A).

We next compared the changes in relative abundance for consecutive timepoints for each bacterial group at genus level, between the two treatment groups. During weeks 1–3 when only bLF was supplemented, the changes in the relative abundances of the bacterium *Holdemanella* were significantly higher in the intervention group (*p* = 0.01) compared to the placebo group (Figure 2A). The increase in the relative abundance of the genus *Holdemanella* was observed only after the first three weeks of intervention, after which it did not change anymore until the end of the study (Figure 2B). During weeks 4–6, when GOS was added along with the bLF supplementation, the change in the relative abundances of the genus *Bifidobacterium* increased significantly (*p* = 0.03) compared to the placebo group (Figure 2C). For the last phase of the study, weeks 7–9 when Vitamin D was also added along with bLF and GOS, no differences were reported in the changes of the relative abundances of the bacterial genus-level taxa for the two groups. Despite the fact that no further increase was recorded for bifidobacteria during the last three weeks of the study, their abundances remained significantly higher compared to the T2 (week 6), when only bLF was supplemented (Figure 2D).

Overall these results suggest that, during the nine weeks of the study, no significant differences were observed at the community level and only specific bacterial taxa were affected upon bLF supplementation and GOS addition.

### 4.1. Fecal Short Chain Fatty Acids

To characterise the metabolic activity of the fecal microbiota, the concentration of the SCFAs acetate, propionate, butyrate, iso-butyrate and succinate was determined in fecal water of all samples. Results showed that acetate, propionate and butyrate were the main SCFAs detected in the fecal samples but they did not differ significantly between the placebo and the intervention group at any of the timepoints or within the groups between the different treatment phases (Figure 3).

### 4.2. Intestinal Inflammatory Markers

To determine if the nutritional intervention had an effect on inflammatory status and barrier function in the subjects, levels of the pro-inflammatory markers calprotectin, and α-1-antitrypsin, and the intestinal barrier marker zonulin were measured in fecal water. No differences in any of these markers were apparent between the two groups at any timepoint. When looking for differences in time within groups, a significant increase over time was detected in zonulin levels (*p* < 0.05) for the placebo group, while no effect was observed for the intervention group (Figure 4).

## 5. Discussion

In the present study, we investigated the effect of bLF, administered alone (first three weeks) or combined with GOS (week 4–9) and vitamin D (week 7–9) on gut microbiota composition, metabolites and intestinal inflammatory status of healthy elderly women, selected from a general population. Results showed that minor effects were observed on overall gut microbiota composition of the intervention group, compared with the placebo. No difference was observed in the levels of any of the fecal SCFAs between the two groups, suggesting that metabolic activity of the gut microbiota was not affected by the supplements. Overall these results suggest that bLF administration, when combined with GOS and vitamin D, is safe for elderly women and does not significantly modify the gut microbiota.

For a period of nine weeks, bLF was supplemented at a dose of 1 g/day, but only during the first three weeks of the study, when only bLF was supplemented, was the relative abundance of the bacterial genus *Holdemanella* increased. This finding suggests that bLF can promote an abundance of *Holdemanella* but the effect is diminished after the addition of GOS. The bacterium *Holdemanella bioformis* is the only described species from the genus *Holdemanella,* and it has been isolated previously from human feces [31]. *H. bioformis* can produce SCFAs, mainly butyrate, but also long chain fatty acids, such as the anti-inflammatory 3-hydroxyoctadecaenoic acid (C18-3OH) [32]. Beneficial effects of this bacterium have been reported in mice studies, where administration of *H. bioformis* isolated from the feces of a metabolically healthy human to obese mice, was able to improve glucose tolerance through the restoration of GLP-1 function and production of C18 fatty acid in the cecum, suggesting that *H. biformis* could be potentially used for management of type-2 diabetes [33]. Nevertheless, since only one bacterium was affected by the bLF supplementation, we can conclude that bLF has a minor effect on the fecal microbiota composition of the elderly. To the best of our knowledge, to date there is only one study available testing for the effect of bLF in adults and elderly and, similarly to our results, no significant changes were reported post-supplementation of 200 mg bLF/day in healthy adults for four weeks [24]. After the two-week washout period, bLF was supplemented again to the same cohort at a dose of 600 mg bLF/day and the authors report minor differences. The minor effects of bLF on the gut microbiota might be explained by the fact that bLF is mainly digested in the small intestine, and only fractions of the bLF reach the colon [34]. Hence, future studies could consider focusing on the effect of bLF on the small intestinal microbiota to better understand how bLF affects the microbiome in the elderly population.

At the end of week 6, after GOS was supplemented combined with bLF to the intervention group for three weeks, the relative abundances of bifidobacteria were increased in the intervention group, in comparison to the placebo group. GOS are well known for their prebiotic effect and are among the main components of infant formula aiming to stimulate growth of intestinal bifidobacteria and to compensate for the absence of human milk oligosaccharides. In both adults [35] and the elderly [36], GOS has been shown to possess a bifidogenic effect, and increases in bifidobacterial abundances have been reported in comparison with placebo group. In line with these findings, in the present study we also observed an increase in bifidobacterial relative abundance after GOS supplementation, and the increase was significantly higher compared to the placebo group. Previous studies, suggested that a GOS dosage of 5 g/day had a bifidogenic effect in healthy adults after three weeks of supplementation, but a dosage of 2.5 g/day did not [37]. In the current study, elderly subjects were supplemented with 3.67 g/day containing 2.64 g/day of pure (active) GOS and we observed the bifidogenic effect. This dosage is the same as the pure GOS contained in 5 g of GOS as described by Vulevic et al. [36], and thus the bifidogenic effect observed in our cohort is in line with this earlier study. Nevertheless, our study is the first to demonstrate that, upon concurrent supplementation of bLF with low levels of GOS, GOS is still able to promote bifidobacteria in the gut of elderly women.

We also examined the subjects’ feces for markers related to gut health and no significant changes were found for calprotectin as marker of gut inflammation [38] and zonulin as marker of gut integrity [39]. The baseline levels for these parameters were all within the normal range, indicating that the subjects did not have gut health issues [39,40]. Interestingly, for the placebo group there was an increasing trend in the levels of fecal zonulin, which reached significance in comparison with baseline levels after timepoint 2. During the first three weeks the subjects received maltodextrin at a total dose of 5 g/day, but after timepoint 3 subjects received an increased dosage of 9 g/day to match the amount of ingested compounds in the treatment group and GOS supplementation. Hence it is possible that the amount of maltodextrin affected zonulin levels in this study, which to the best of our knowledge has not been recorded previously in the literature. Maltodextrin is generally considered as safe, but some recent evidence supports a potential pro-inflammatory effect for a specific maltodextrin in the murine gut [41], although there is no evidence to the best of our knowledge for such activity in humans.

Limitations of our study include the small sample size, and further clinical trials with bigger cohorts will be necessary to verify our findings. Moreover, due to the variation in diet in the general population, inclusion of a food frequency questionnaire is highly recommended, to account for habitual diet. For example, information on fiber consumption is an important parameter for gut microbiota composition. Finally, since bLF is mainly digested in the small intestine the effects on the small intestinal microbiota should be considered in future studies.

We previously reported that the supplementation of bLF in this intervention study enhanced innate immune responses of plasmacytoid dendritic cells to TLR7 stimulation in elderly women, suggesting enhanced anti-viral immune function [7]. Overall, the results from this study show that supplementation of bLF increased the relative abundance of the genus *Holdemanella* in the fecal microbiota of healthy elderly women, and further addition of GOS enhanced the relative abundance of bifidobacteria. Moreover, bLF supplementation did not have an adverse effect on intestinal health based on the gut markers measured and did not affect prebiotic activity of GOS.

## Figures and Tables

**Figure 1 nutrients-14-02468-f001:**
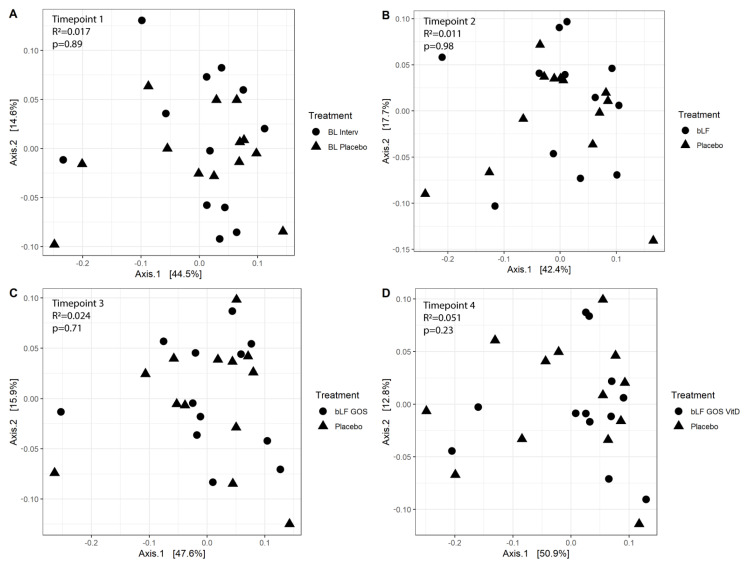
Principal coordinates analysis within each timepoint, based on weighted UniFrac distances. (**A**) At baseline between the intervention (BL Interv) and placebo (BL Placebo) group. (**B**) After 3 weeks, when the intervention group was receiving bovine lactoferrin (bLF). (**C**) After 6 weeks, when the intervention group was receiving bovine lactoferrin and galactoligosacharides (bLF GOS). (**D**) After 9 weeks, when the intervention group was receiving bovine lactoferrin, galactoligosacharides and vitamin D (bLF GOS VitD). No differences were observed between the two groups as determined by ADONIS analysis of variance with significance set at *p* < 0.05. Shapes represent the treatment each subject received at the respective timepoint. Percentages at the axes indicate the fraction of observed variation explained.

**Figure 2 nutrients-14-02468-f002:**
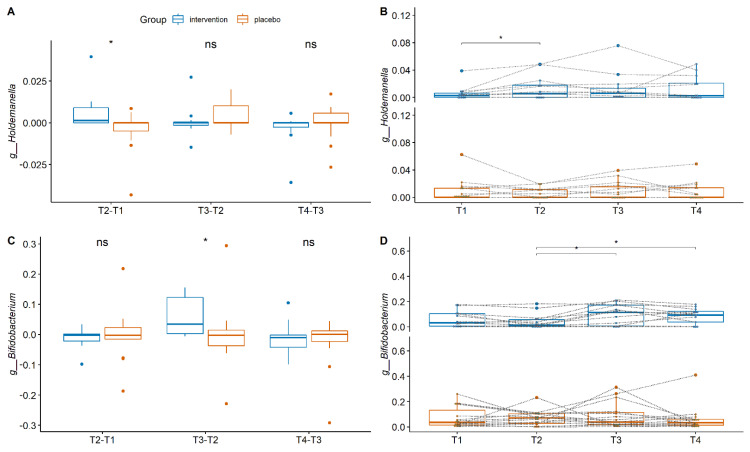
Bacterial taxa that are differentially abundant between the two groups. (**A**) Differences in relative abundances between two consecutive timepoints for the genus *Holdemanella*. (**B**) The relative abundances of the genus *Holdemanella* through time within the two groups. (**C**) Differences in relative abundances between two consecutive timepoints for the genus *Bifidobacterium*. (**D**) The relative abundances of the genus *Bifidobacterium* through time within the two groups. The boundaries of the box on the bottom indicate the 25th percentile, and the line within the box represents the median, and the boundary of the box on the top the 75th percentile. For panels (**A**,**C**), the differences between groups were calculated using Mann-Whitney test. For panels (**B**,**D**), the differences through time were calculated using the paired Wilcoxon Rank test, and data from individual participants are connected by dotted lines. Stars above bars represent statistical differences compared to the time-matched control samples (* *p* < 0.05).

**Figure 3 nutrients-14-02468-f003:**
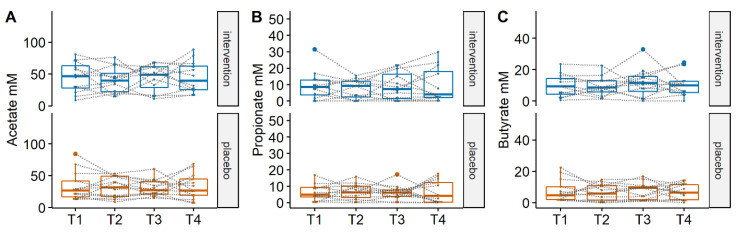
Concentrations of the SCFAs (**A**) acetate, (**B**) propionate and (**C**) butyrate detected in the fecal water of the samples in the two groups. No significant differences were detected between the two groups. The boundaries of the box on the bottom indicate the 25th percentile, the line within the box represents the median, and the boundary of the box on the top the 75th percentile, and data from individual participants are connected by dotted lines. Differences between groups and within groups were calculated using Linear mixed effects models.

**Figure 4 nutrients-14-02468-f004:**
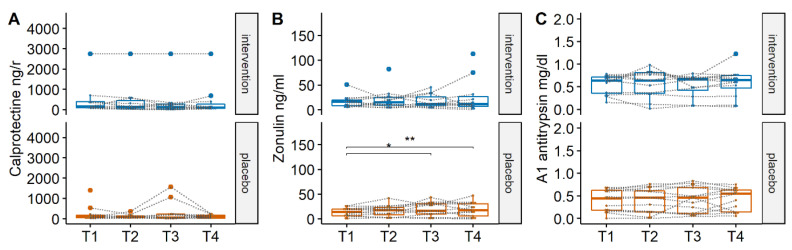
Concentrations of (**A**) calprotectin, (**B**) zonulin and (**C**) antitrypsin detected in the fecal water of the samples between the two groups. No significant differences were detected between the two groups in any of the markers tested. *, ** A significant difference was detected only for zonulin which increased in the placebo group. The boundaries of the box on the bottom indicate the 25th percentile, the line within the box represents the median, and the boundary of the box on the top the 75th percentile, and data from individual participants are connected by dotted lines. Differences between groups and within groups were calculated using Linear mixed effects models.

**Table 1 nutrients-14-02468-t001:** Baseline characteristics of the participants.

	Total Group	Intervention Group	Placebo Group
Number of subjects	*n* = 25	*n* = 12	*n* = 13
Age: median (range) 74.5 (69–85) 74 (70–84) 76 (69–85)	74 (69–85)	74 (70–84)	74 (69–85)
BMI: median (range)	24.5(20.3–29.4)	23.2 (20.3–29.0)	24.7 (20.8–29.4)
Reported arthrosis: number Y/N	6/25	3/12	3/13
Use of vitamin D before the study:	9/25	5/12	4/13
Medication blood pressure/cholesterol:	10/25	5/12	5/13

**Table 2 nutrients-14-02468-t002:** PERMANOVA results at the different timepoints.

	T1	T2	T3	T4
Age	R_2_ = 0.035	R_2_ = 0.028	R_2_ = 0.018	R_2_ = 0.017
	*p*-value = 0.47	*p*-value = 0.67	*p*-value = 0.86	*p*-value = 0.87
BMI	R_2_ = 0.033	R_2_ = 0.036	R_2_ = 0.024	R_2_ = 0.011
	*p*-value = 0.51	*p*-value = 0.47	*p*-value = 0.71	*p*-value = 0.97
Medication	R_2_ = 0.032	R_2_ = 0.046	R_2_ = 0.061	R_2_ = 0.064
	*p*-value = 0.58	*p*-value = 0.31	*p*-value = 0.93	*p*-value = 0.17
Osteoarthritis	R_2_ = 0.074	R_2_ = 0.086	R_2_ = 0.064	R_2_ = 0.059
	*p*-value = 0.09	*p*-value = 0.07	*p*-value = 0.67	*p*-value = 0.2
Use of vitamin D before the study	R_2_ = 0.017	R_2_ = 0.024	R_2_ = 0.025	R_2_ = 0.063
	*p*-value = 0.91	*p*-value = 0.75	*p*-value = 0.69	*p*-value = 0.17

## Data Availability

The sequencing data are available at the European Nucleotide Archive with accession number PRJEB52848.

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
