# Peer review of "The Effect of Nutritional Intervention with Lactoferrin, Galactooligosacharides and Vitamin D on the Gut Microbiota Composition of Healthy Elderly Women"

_nutrients, 2022, doi:10.3390/nu14122468_

Round 1
Reviewer 1 Report
Thank you for the opportunity to revise this manuscript and many thanks to the authors for this interesting study. Lactoferrin can modulate both the host immunity and gut microbiota.
However, clinical studies on the effects of the intervention on gut microbiota composition and function and exploring the effect on gut health based on intestinal inflammatory and barrier markers are lacking.
In general, the study gives an interesting multifactorial impression of the different aspects of the lactoferrin on gut microbiota
However, I suggest some improvements.
The introduction could be expanded with other studies that investigated the role of lactoferrin and its immunomodulatory effects in relationship with viruses and bacteria, especially in view of the pandemic situation. I suggest the following references “Sinopoli, A., Isonne, C., Santoro, M. M., & Baccolini, V. (2022). The effects of orally administered lactoferrin in the prevention and management of viral infections: A systematic review. Reviews in medical virology, 32(1), e2261”; “Sienkiewicz, M., JaÅ›kiewicz, A., Tarasiuk, A., & Fichna, J. (2021). Lactoferrin: An overview of its main functions, immunomodulatory and antimicrobial role, and clinical significance. Critical Reviews in Food Science and Nutrition, 1-18”.
Methods are well structured.
The discussion is unfocused. I suggest synthesizing it into three parts starting from the results obtained.
In addition, it’s very important to underline the limits of this study. One of these could be that the sample considered is very small.
Reviewer 2 Report
Thank you very much for letting me review this interesting article by Konstanti et al.
The article is a study on the fecal microbiota of subjects already enrolled in another study, published by the same team, and aims to investigate the prebiotic role of lactoferrin, alone or in association with GOS and vitamin D.
The rationale is noteworthy, the work well-conducted, methods and results are well-described.
The authors concluded that lactoferrin supplementation may increase the amount of Holdemanella, and GOS may increase the relative abundances of Bifidobacterium strains. No evidence of an increase in SCFA in the intervention group has been reported compared to the placebo.
My main concern is about the role of individual diet on gut microbiota. In the work, there is no described what is the "habitual diet" of patients (it is only written that subjects were instructed to maintain their habitual diet). Since the diet is the major driver of gut microbiota (in people who don't get antibiotics or probiotics), this is a crucial point at the base of every other inference. There is no mention of the amount of fiber (and hence prebiotic fiber) already consumed by patients in their habitual diet.
In my opinion, a common diet should have been given before and during the intervention; alternatively, a food frequency questionnaire (FFQ) collected before starting supplementation and thereafter, should have been administered to compare groups and avoid bias related to individual dietary patterns or ongoing changes.
In case the Authors do not have such information, this limit should be highlighted and discussed.
Minor remarks:
- abstract is too long. Please, reduce it to 250-300 words.
- materials and methods (line 120): change 800 IE to 800 IU
- The caption of Figure 1 is not isolated from the main text.
Round 2
Reviewer 2 Report
The Authors addressed all my concerns. I have no further questions.